# Construction of Hollow Co_3_O_4_@ZnIn_2_S_4_ p-n Heterojunctions for Highly Efficient Photocatalytic Hydrogen Production

**DOI:** 10.3390/nano13040758

**Published:** 2023-02-17

**Authors:** Zijian Xin, Haizhao Zheng, Juncheng Hu

**Affiliations:** Hubei Key Laboratory of Catalysis and Materials Science, School of Chemistry and Materials Science, South-Central Minzu University, Wuhan 430074, China

**Keywords:** ZIF-67, hollow Co_3_O_4_ nanocages, ZnIn_2_S_4_ nanosheets, p-n heterojunction, photocatalytic hydrogen production, visible light

## Abstract

Photocatalysts derived from semiconductor heterojunctions for water splitting have bright prospects in solar energy conversion. Here, a Co_3_O_4_@ZIS p-n heterojunction was successfully created by developing two-dimensional ZnIn_2_S_4_ on ZIF-67-derived hollow Co_3_O_4_ nanocages, realizing efficient spatial separation of the electron-hole pair. Moreover, the black hollow structure of Co_3_O_4_ considerably increases the range of light absorption and the light utilization efficiency of the heterojunction avoids the agglomeration of ZnIn_2_S_4_ nanosheets and further improves the hydrogen generation rate of the material. The obtained Co_3_O_4_(20) @ZIS showed excellent photocatalytic H_2_ activity of 5.38 mmol g^−1^·h^−1^ under simulated solar light, which was seven times more than that of pure ZnIn_2_S_4_. Therefore, these kinds of constructions of hollow p-n heterojunctions have a positive prospect in solar energy conversion fields.

## 1. Introduction

The depletion of conventional fossil fuels and the environmental climate issues are forcing people to search for renewable energy [1,2,3,4,5]. Hydrogen, as a renewable energy source, possesses abundant reserves, powerful chemical bonding energy, and environmentally friendly features [6]. For these reasons, photocatalytic hydrogen production from water has attracted widespread attention as an effective way to convert solar energy to clean energy [7]. Honda and Fujishima published ground-breaking research on photoelectrocatalytic water splitting on a TiO_2_ electrode in 1974 [8]. Since then, many semiconductor materials have evolved, including metal oxides [9,10], metal sulfides [11], bismuth halides [12,13], and g-C_3_N_4_. [14]

As one of the typical n-type semiconductor photocatalytic materials, ZnIn_2_S_4_, benefiting from a suitable band gap (*Eg* ≈ 2.7 eV) and energy band position, has been widely used in photocatalytic hydrogen production [15]. Furthermore, 2D ZnIn_2_S_4_ nanosheets have the advantages of large specific surface areas and abundant surface-active sites, which facilitate involvement in a wide range of photocatalytic redox reactions (for instance, photocatalytic H_2_ evolution, Cr^VI^ reduction, and CO_2_ conversion) [16]. But the carrier recombination of single ZnIn_2_S_4_ nanosheets is still severe. They tend to agglomerate into nanospheres, which vastly reduces their specific surface area and the quantity of active sites, and leads to more disordered carrier migration [17,18]. Previous studies have revealed that coupling ZnIn_2_S_4_ with a semiconductor to build a heterojunction can enhance their performance, and mitigate photo corrosion and carrier compounding. Li et al. placed In(OH)_3_ in sheet planes along the edges of ZnIn_2_S_4_ nanoplates. Under light irradiation, the photocatalytic H_2_ evolution performance was about 4.9 times higher compared with pristine ZnIn_2_S_4_, which effectively separates and transfers charges [19]. Liu et al. reported a hybrid photocatalyst prepared by coupling two-dimensional ZnIn_2_S_4_ nanosheets with amino-functionalized Ti-based MOF [14]. In another study by Zuo and colleagues, sandwich-like hierarchical MXene-ZnIn_2_S_4_ heterostructures were successfully produced by anchoring ultrathin ZnIn_2_S_4_ nanosheets on two surfaces of Ti_3_C_2_T_X_MXene [20].

As a member of p-type semiconductors, Co_3_O_4_ has many advantages, including its strong chemical stability and good electrical, magnetic, and catalytic properties. Thus, Co_3_O_4_ was used as an oxidative cocatalyst to effectively capture holes generated by the photocatalytic host and promote effective spatial separation of carriers. For instance, Wang and colleagues hydrothermally synthesized 0D/1D TiO_2_/Co_3_O_4_ p-n heterojunctions. The built-in electric field formed in the p-n heterojunctions and Co_3_O_4_ co-catalyst effect synergistically enhanced carrier transfer and spatial separation [21]. Liu and colleagues made the in situ growth of ZIF-67 onto g-C_3_N_4_. The p-n heterojunction g-C_3_N_4_@Co_3_O_4_ was synthesized by low-temperature sintering. The NO degradation efficiency of the best proportion of the composite sample reached 57% [22]. In addition, the hollow structure has a significant influence the photocatalytic performance.

The hollow structure has drawn increasing attention due to the large surface area and plentiful reactive sites, favoring adsorption activation of the reaction substrate and therefore promoting the redox reaction. Compared with the solid structure, the hollow structure has a thin shell, which makes the carrier transfer pathway shorter and the carrier recombination weaker. The hollow structure can also cause light to be reflected and scattered inside the material, improving light absorption and usage [23,24,25].

Herein, we successfully fabricated a ZnIn_2_S_4_@Co_3_O_4_ heterojunction by loading ultrathin two-dimensional ZnIn_2_S_4_ nanosheets onto hollow dodecahedral Co_3_O_4_ nanocages derived from ZIF-67 by an oil bath. The constructed ZnIn_2_S_4_@Co_3_O_4_ p-n heterojunction effectively hinders electron-hole recombination. Additionally, the dark hollow structure of Co_3_O_4_ significantly expands the spectrum of light absorption and the heterojunction’s ability to use light efficiently by preventing the accumulation of ZnIn_2_S_4_ nanosheets. Consequently, the optimal ZnIn_2_S_4_@Co_3_O_4_ composites show apparent enhancement in H_2_ production efficiency compared with pure ZnIn_2_S_4_. The ZnIn_2_S_4_@Co_3_O_4_ heterostructure is stable within 12 h.

## 2. Experimental Section

### 2.1. Chemicals and Materials

All chemical reagents used in this experiment were purchased from Sinopharm Chemical Reagent Co., Ltd. (Shanghai, China), which include 2-methylimidazole(C_4_H_6_N_2_), indium nitrate tetrahydrate (InCl_3_·4H_2_O), cobalt nitrate hexahydrate(Co(NO_3_)_2_·6H_2_O), anhydrous zinc chloride (ZnCl_2_), thioacetamide (C_2_H_5_NS), glycerol (C_3_H_8_O_3_), hydrochloric acid (HCl). All of them were analytical grades, without further purification.

### 2.2. Synthesis of ZIF-67

First, 2.91 g of Co(NO_3_)_2_·6H_2_O and 3.28 g of 2-methylimidazole were dissolved in 200 mL of methanol solution respectively and then the former was slowly poured into the latter with stirring at room temperature for 24 h. The obtained was centrifuged and washed three times with methanol, and then dried overnight at 60° in a vacuum oven.

### 2.3. Synthesis of Co_3_O_4_

A certain amount of synthesized ZIF-67 was laid flat in a crucible, placed in a muffle furnace, and calcined at a 2 °C/min rate for 2 h under an air atmosphere.

### 2.4. Synthesis of ZnIn_2_S_4_@Co_3_O_4_

A certain amount of as-prepared Co_3_O_4_ powder was dispersed into the mixture of glycerol (8 mL) and distilled water (32 mL) with the aid of ultrasonication. Then, ZnCl_2_ (136.3 mg), C_2_H_5_NS (293.24 mg), and InCl_3_·4H_2_O (293.24 mg) were added into the mixture and ultrasonicated for 5 min and stirred for 25 min. Next, the obtained suspension was transferred into a 100 mL flask and maintained at 80 °C for 2 h. The resultant solid products were washed with distilled water and ethanol several times and dried at 60 °C. The as-synthesized ZnIn_2_S_4_@Co_3_O_4_ samples with 10 mg, 20 mg, and 40 mg of Co_3_O_4_ were labeled as Co_3_O_4_@ZIS(10), Co_3_O_4_@ZIS(20), and Co_3_O_4_@ZIS(40), respectively.

### 2.5. Characterization

The crystallinity of the samples was investigated by X-ray diffraction (XRD) (Bruker D8 Advance; Cu Kα = 1.5404Å,40 kV, 40 mA;) with a scanning rate of 0.05°/s, using Bragg measurements, and the samples were in powder form, prepared by pressing the slices. The morphologies and sizes of the samples were characterized by SU8010 field-emission scanning electron microscope (FESEM, Hitachi, Japan) with secondary electron measurements. The beam energy was 10μA, and the samples were fixed with liquid conductive adhesive in deceleration mode with an acceleration voltage of 4.5 kV and a landing voltage of 2.0 kV. Transmission electron microscope (TEM) and high-resolution transmission electron microscopy (HRTEM) were measured on a Talos F200X microscope which operated on an accelerating mode of 200 kV. Samples were prepared by crushing and sieving and then dispersed with solvent onto a copper mesh carbon film. The carbon film was then fixed on the sample rod by clamping the carbon film. The measurement methods include bright field, dark field, and HAADF. X-ray photoelectron spectroscopy (XPS) (Thermo Scientific ESCALAB Xi^+^) was carried out to analyze the chemical states of the samples. The X-ray source is monochromic Al Kα radiation (hν = 1486.6 eV). The samples were in powder form. Peak deconvolution and spectral analysis were conducted using xpspeak41 software. The background was chosen as smart and the fitted line pattern was L/G Mix. Linewidth differences were relatively subtle. Semiquantitative analysis of elemental content was performed using the Thermo Scientific ESCALAB Xi+ self-contained database, and a corrected Scofield sensitivity factor. The UV–Vis diffused reflectance spectra (DRS) were recorded using a Cary Series UV–Vis–NIR spectrophotometer (Agilent Technologies). Photoluminescence (PL) measurements were characterized on a Hitachi F-7000 with a 150 W Xe lamp.

### 2.6. Photocatalytic Activities Test

In the photocatalytic activity test system, 20 mg of catalyst was dissolved in a 100 mL aqueous triethanolamine solution (V_TEOA_:V_H2O_) and placed in a 400 mL quartz reactor. Then, nitrogen was infused to remove the air from the inner cavity of the reactor for 30 min upon stirring. After the air was exhausted, the reactor was irradiated with a 300 W Xenon lamp equipped with a 420 nm filter simulating visible light, and circulating water was passed through the outer layer of the reactor to avoid the existence of thermal catalysis. After the hydrogen evolution reaction was performed, the nitrogen and hydrogen mixture gas in the 0.4 mL reactor was taken out by the injection needle and taken every 30 min for a total of six times. Hydrogen content was detected by gas chromatography (FULI 9790) equipped with an AE.5A molecular sieve column (3 m × 3 mm) and the oven temperature, thermal conductivity (detector) temperature, and injector temperature of the gas phase are set to 80 °C, 120 °C, and 100 °C, respectively.

### 2.7. Catalyst Photoelectric Performance Test

In this work, we carried out two photoelectric tests: photocurrent and electrochemical impedance. A three-electrode system and a Chi-760E Chenhua electrochemical workstation are used for the photoelectric test. The three-electrode system used 0.5 mol/L Na_2_SO_4_ solution as the electrolyte, platinum electrode as the counter electrode, Ag/AgCl electrode as the reference electrode, and ITO conductive glass coated with a layer of the sample as the working electrode.

## 3. Results and Discussion

At room temperature, Zn^2+^ and 2-methylimidazole synthesize regular dodecahedral ZIF-67 in a methanol solution, which is subsequently placed in a muffle furnace for calcination. The organic ligand in ZIF-67 will undergo oxidation, forming hollow, regular dodecahedrons made of Co_3_O_4_ particles, and other gases will be liberated. The hollow regular dodecahedron Co_3_O_4_ will undergo a half-hour ultrasonic stir with sulfur, indium, zinc, and glycerol, followed by a two-hour immersion in oil at 80 °C in a round-bottled flask. This results in the hollow Co_3_O_4_ surface homogeneous load with extremely thin ZnIn_2_S_4_ heterojunction material.

Figure 1 shows the proposed synthetic method for the p-n heterojunction of ZnIn_2_S_4_@Co_3_O_4_. ZIF-67 was oxidized into a hollow dodecahedral structure composed of Co_3_O_4_ particles. Then, the heterojunction material with ultrathin ZnIn_2_S_4_ uniformly loaded on the surface of hollow Co_3_O_4_ was obtained.

XRD is used to analyze the crystal phase structure of materials. As shown in Figure 2a, the precursor ZIF-67 is effectively synthesized, which is compatible with previously published research [26]. In Figure 2b, the peaks of pure Co_3_O_4_ at 19.05°,31.25°, 36.80°, 44.78°, 59.42°, and 65.24° corresponding to the (111), (220), (311), (222), (511), (440) planes (JCPDS:73-1701) respectively [22]. The positions of the three strong diffraction peaks of ZnIn_2_S_4_ at 21.63°, 27.45°, and 47.49° respectively, correspond to the (006), (102), and (110) crystal planes of ZnIn_2_S_4_ (JCPDS:65-2023) [15]. For Co_3_O_4_@ZIS, the phase of Co_3_O_4_ and ZIS are all detected. Compared with the pure sample, they all show the same diffraction peak with no shift, which indicates the successful formation of Co_3_O_4_@ZIS heterojunction.

The morphology of the obtained resultant products is examined by FE-SEM. A hollow structure of Co_3_O_4_ with a 500 nm size without collapse and fragmentation in Figure 3(a1–a3) was obtained. The spherical ZnIn_2_S_4_ comprises many sheets with a diameter of about 1 μm. Figure 3(b1–d3) show SEM images of composite samples Co_3_O_4_ (10) @ ZIS, Co_3_O_4_ (20) @ ZIS and Co_3_O_4_ (40) @ ZIS with different proportions, respectively. In terms of surface morphology, these composite samples are spherical, however, the 2D nanosheet arrangement on their surfaces is more disordered than that of pure ZnIn_2_S_4_ microspheres. The particle sizes of pure ZnIn_2_S_4_ flower-like microspheres are quite different from those of all composite samples, the size of pure ZnIn_2_S_4_ spheroids is about 1μm, and that of composite Co_3_O_4_ (X) @ ZIS is less than 1 μm (Figure 3(e1–e3)). With the increase of the proportion of Co_3_O_4_ added in the synthesis process, the particle size of the spherical composite decreases continuously, which probably results from the dispersion of ZnIn_2_S_4_ nanosheets with the same mass on more Co_3_O_4_ dodecahedron, which causes the decrease in particle size for the spherical composite, indicating the successful construction of heterojunction from the side.

The microstructure of the sample is analyzed by STEM. The pure ZnIn_2_S_4_ is a solid flower sphere structure (Figure 4a–c), and the dodecahedral structure of Co_3_O_4_ is composed of many Co_3_O_4_ particles Figure 4i–k, which corresponds to the previous SEM results. The hollow structure of Co_3_O_4_ (20)@ZIS is significantly complete and obvious in Figure 4e–g. Furthermore, in the HRTEM image of ZnIn_2_S_4_ in Figure 4d, lattice distances are 0.190 nm, 0.287 nm, and 0.32 nm, respectively, corresponding to the three crystal planes (110), (104), and (102) of ZnIn_2_S_4_ [27]. Three kinds of lattice fringes of Co_3_O_4_ were measured. The lattice distances were 0.284 nm, 0.245 nm, and 0.202 nm corresponding to its three crystal planes (220), (311), and (400), respectively [15]. In addition, the HRTEM image of the edge of the Co_3_O_4_ (20) @ ZIS sample shows that two lattice distances (Figure 4h) are 0.323 nm and 0.283 nm, respectively, corresponding to the (102) and (104) crystal planes of ZnIn_2_S_4_, which proves that ZnIn_2_S_4_ exists in the composite sample. Since Co_3_O_4_ is completely covered by ZnIn_2_S_4_, the crystal surface of Co_3_O_4_ cannot be observed.

To further confirm the intimate contact interface element between Co_3_O_4_ and ZnIn_2_S_4_, the HAADF-STEM-EDX was undertaken. Figure 4m demonstrates the uniformly distributed nature of Zn, S, In, Co, and O. From the element distribution of Co and O, it is apparent that Co_3_O_4_ is composed of particles, which is consistent with the findings in previous SEM and TEM diagrams. The successful synthesis of the composite sample was further verified.

The X-ray photoelectron spectroscopy (XPS) analysis of the Co_3_O_4_ @ ZIS, Co_3_O_4,_ and ZnIn_2_S_4_ composites is shown in Figure 5a. As shown in Figure 5b, the Co 2p spectrum is composed of two spin-orbit doublets and two satellite peaks. The binding energies of the first peak in the composite Co_3_O_4_(20)@ZIS are at 779.70 eV and 781.36 eV, and the second peaks are at 794.28 eV and 796.03 eV, corresponding to the Co 2p_3/2_ and Co 2p_1/2_ orbitals, respectively, which indicate the coexistence of Co^2+^ and Co^3+^ [28,29]. The high-resolution XPS spectrum of the O element shows three peaks at 529.65 eV, 531.39 eV, and 532.24 eV in Figure 5c. The first and second characteristic peaks correspond to lattice oxygen in metal oxides and hydroxyl radicals on the surface of materials, while the latter corresponds to adsorbed water on the surface [30,31,32,33,34]. The peaks (Figure 5d) at 1021.73 and 1044.85 eV are attributed to Zn 2p_3/2_ and Zn 2p_1/2_, respectively, indicating the presence of Zn^2+^ [33]. Figure 5e shows that the two peaks at 444.62 eV and 452.21 eV are assigned to In 3d_5/2_In and In 3d_3/2_ respectively, indicating that the chemical state of the cation in the composite is plus three [18]. S 2p spectrum (Figure 5f) can be resolved into two peaks including 162.8 and 161.6 eV, corresponding to the S 2p_1/2_ and S 2p_3/2_ orbitals of S^2-^ [34]. Overall, the XPS results indicate that the binding energies of the Zn, In, and S have a level of red shift compared with pure ZnIn_2_S_4_, which confirms the successful construction of the heterojunction [35].

The samples’ nitrogen adsorption–desorption and pore size distribution curves are shown in Figure 6a–c. Naturally, type IV isotherm and type H3 hysteresis rings are found in all samples, demonstrating the presence of capillary condensation. The materials’ microstructure and specific surface area are typically closely connected. As can be observed from the little figure, the three samples all have pore sizes that are mostly spread between 1 and 75 nm, which indicates that they are all mesoporous materials. In Table 1, Co_3_O_4_(20)@ZIS, ZnIn_2_S_4_, and Co_3_O_4_ had specific surface area distributions of 121, 15, and 71 m^2^ g^−1^, respectively. Evidently, the specific surface area and pore volume of the Co_3_O_4_(20)@ZIS composite samples increased with the addition of ZnIn_2_S_4_ in comparison to pure Co_3_O_4_. It is possible that the composite catalyst will offer more active centers, which will make it easier for photocatalytic hydrogen evolution events to occur. Another angle is that the success of the catalyst preparation is demonstrated by the modification of the adsorption characteristics of the composite samples.

The characterization of phase, morphology, and structure proves the successful synthesis of hollow structure Co_3_O_4_, and ZnIn_2_S_4_ successfully grows on the surface of Co_3_O_4_. Based on this point, the hydrogen production performance of all samples is tested under visible light in Figure 7a. Co_3_O_4_ (20) @ ZIS represents the best ration of the composite sample. Its hydrogen production is as high as 16.14 mmol g^−1^ in three hours, while the hydrogen production performance of pure ZnIn_2_S_4_ is only 2.22 mmol g^−1^. In addition, the hydrogen production of Co_3_O_4_ is almost 0 mmol g^−1^, which may be due to the narrow band gap of Co_3_O_4_, which causes the carriers to recombine rapidly and prevents them from moving to the surface to take part in the proton reduction process. Figure 7b clearly shows the hydrogen production rate of samples with different proportions, the Co_3_O_4_ (20) @ ZIS with the best proportion reaches 5.38 mmol g^−1^ L^−1^. In addition, the hydrogen production rate first rises and then falls with the increase of the proportion of ZnIn_2_S_4_ in the composite sample, which may be due to too much ZnIn_2_S_4_ being loaded onto the surface of the Co_3_O_4_, thus making the active sites on Co_3_O_4_ not fully exposed, and resulting in the hole not reacting with sacrificial agents in time, the increase of carrier recombination efficiency, and the decrease of hydrogen production performance [36]. Subsequently, the hydrogen production of Co_3_O_4_ (20) @ ZIS is tested for a long time to test the stability of the sample. Meanwhile, as shown in Figure 7c and Appendix A, the hydrogen production rate is essentially constant within 12 h, proving the stability of the sample. It also explains that the loading of Co_3_O_4_ enhances the photocatalytic stability of ZnIn_2_S_4_. Co_3_O_4_ effectively captures the photo-generated holes produced by the photoexcitation of ZnIn_2_S_4_, which significantly reduces the oxidation process of S^2−^ by holes, and effectively alleviates the photo-corrosion problem of ZnIn_2_S_4_.

To find out the potential reasons for the improved performance of Co_3_O_4_ (X) @ ZIS, a solid-ultraviolet diffuse reflectance (DRS) test, photoluminescence emission spectrum (PL), photocurrent, electrochemical impedance, and other photoelectrochemical tests are carried out to evaluate the photoelectric performance, carrier separation, and transfer efficiency of the samples. Figure 8a is the solid-ultraviolet diffuse reflectance spectrum (UV-VIS DRS). The absorption range of ZnIn_2_S_4_ is about 510 nm, and Co_3_O_4_ absorbs in the wavelength range of ultraviolet, visible, and near-infrared light. With the increase of the proportion of Co_3_O_4_, the absorption range of the composite sample is red-shifted gradually, which corresponds to the color variation of the sample, indicating that ZnIn_2_S_4_ is uniformly loaded on the surface of the Co_3_O_4_ dodecahedron with a hollow structure and the existence of black Co_3_O_4_ increases the absorption range and intensity of the sample. According to the formula:(1)$$\alpha h\nu =A{\left(h\nu -E_{g}\right)}^{n/2}$$

The energy band gaps of Co_3_O_4_ and ZnIn_2_S_4_ are calculated to be 1.32 eV and 2.70 eV based on the optical absorption band. This result will be helpful to the later energy band mechanism.

Furthermore, Figure 8b is the PL spectra of all the samples, the emission peaks of Co_3_O_4_ and ZnIn_2_S_4_, and the three composite samples located near 395 nm. Their peak intensities are by Co_3_O_4_ (20) @ ZIS < Co_3_O_4_ (10) @ ZIS < Co_3_O_4_ (40) @ ZIS < ZnIn_2_S_4_, which is consistent with the trend of hydrogen production properties mentioned earlier. The significantly increased separation efficiency of photogenerated electron-hole pairs in the hybrid system is demonstrated by the large quenched PL of all composites compared to Co_3_O_4_ and ZnIn_2_S_4_.

Transient photocurrent response (TPR) in a classical three-electrode system is used to continue the exploration of the photoelectrochemical characteristics of Co_3_O_4_ (X) @ ZIS, to understand the potential reasons for the enhanced carrier separating and transfer ability of Co_3_O_4_ (X) @ ZIS. As shown in Figure 8c, since the black Co_3_O_4_ has a narrow band gap (Eg = 1.32 eV) and belongs to a narrow band gap transition metal semiconductor, the carrier recombination of Co_3_O_4_ is too fast, resulting in no photocurrent signal or its weak signal. Meanwhile, it is found that pure ZnIn_2_S_4_ nanospheres exhibit a small photocurrent intensity due to the corrosion of ternary sulfide and serious carrier recombination caused by the solid structure. As anticipated, after loading ZnIn_2_S_4_ onto the hollow nanostructure of Co_3_O_4_, the photocurrent intensity of the composite sample was significantly higher than that of the two substrates. Among all the composite samples, Co_3_O_4_ (20) @ ZIS has the strongest photocurrent signal, indicating that Co_3_O_4_ in the heterojunction system can make ZnIn_2_S_4_ carriers excited under visible light separate and migrate rapidly. The transfer impedance of photoexcited carriers in the heterojunction is observed via electrochemical impedance spectroscopy (EIS). As shown in Figure 8d, the semicircle formed by the Nernst curve of Co_3_O_4_ (20) @ ZIS is the smallest semicircle in the produced samples, offering the lowest transfer impedance during photocatalytic activity [37,38]. It is in perfect agreement with the photocurrent test and fluorescence test.

Mott–Schottky tests were performed to analyze the energy band structure of Co_3_O_4_ and ZnIn_2_S_4_, and reasonable assumptions were made on the catalytic mechanism of catalyst materials based on the energy band structure (Figure 9). According to the formula:(2)$$\frac{1}{C^{2}}=\frac{2}{\varepsilon \varepsilon_{0}{\mathrm{eN}}_{D}}(V-V_{\mathrm{FB}}-\frac{K_{B}T}{e})$$

The Mott–Schottky curve of ZnIn_2_S_4_, in which the slope of the linear part is positive, indicates that ZnIn_2_S_4_ is an n-type semiconductor, and the flat band potential (E_fb_) is −0.54 eV (−0.76 eV vs. Ag/AgCl), relative to the standard hydrogen electrode. Figure 9b shows the Mott–Schottky curve of Co_3_O_4_, in which the slope of the linear part is negative, indicating that Co_3_O_4_ is a p-type semiconductor, and the flat band potential (E_fb_) is 1.92 eV (1.70 vs. Ag/AgCl) relative to the standard hydrogen electrode. According to earlier findings, the conduction band value for n-type semiconductors is 0.1 eV lower than the flat-band potential, while the valence band value for p-type semiconductors is 0.1 eV higher than the flat-band potential, resulting in ECB (ZnIn_2_S_4_) =−0.64 eV and EVB (Co_3_O_4_) = 2.02 eV. In conjunction with the band gap values determined using the DRS pattern from the prior research, as shown in Figure 9c, Eg (ZnIn_2_S_4_) = 2.70 eV and Eg (Co_3_O_4_) = 1.32 eV, it can be calculated that EVB (ZnIn_2_S_4_) =−2.06 eV and ECB (Co_3_O_4_) = 0.70 eV. This energy band structure promotes the formation of Co_3_O_4_ (X) @ ZIS p-n heterojunction.

The mechanism hypothesis of Co_3_O_4_ (X) @ ZnIn_2_S_4_ heterojunction participating in photocatalytic hydrogen production is shown in Figure 10. Before the Co_3_O_4_ and ZnIn_2_S_4_ contact, ZnIn_2_S_4_ has a higher Fermi level (Ef) than Co_3_O_4_. When ZnIn_2_S_4_ is loaded on Co_3_O_4_, a close contact surface will be formed between them. The negative charge in ZnIn_2_S_4_ is transferred to Co_3_O_4_ through the interface, and the positive charge in Co_3_O_4_ is transferred to ZnIn_2_S_4_ through the interface until their Fermi levels are equal, and a built-in electric field from ZnIn_2_S_4_ to Co_3_O_4_ is formed. It is worth mentioning that the energy band positions of the two will also move with the Fermi level, and the conduction band of Co_3_O_4_ will move to a more negative position than that of ZnIn_2_S_4_ so that electrons on the Co_3_O_4_ conduction band can transfer to ZnIn_2_S_4_ conduction band. However, the built-in electric field can further guide the electron transfer from the conduction band of p-type semiconductor Co_3_O_4_ to the conduction band of ZnIn_2_S_4_, and the hole transfer from the valence band of n-ZnIn_2_S_4_ to the valence band of Co_3_O_4_, which promotes the spatial separation of electron-hole pairs. Therefore, more electrons are transferred to ZnIn_2_S_4_ nanosheets and participate in the proton reduction reaction.

## 4. Conclusions

The nanocomposite Co_3_O_4_@ ZnIn_2_S_4_ with intimate contact was synthesized by the mild oil bath. The p-n heterojunctions were successfully constructed, forming the internal electric field and effectively realizing spatial separation of carriers. Moreover, Co_3_O_4_ broadened the absorbance range and enhanced the absorbance intensity of the composite Co_3_O_4_@ ZnIn_2_S_4_ in this hybridized system. Furthermore, the hollow structure of Co_3_O_4_ further improved the utilization rate of light. As a result, the optimal Co_3_O_4_ (20)@ ZnIn_2_S_4_ photocatalyst exhibited an H_2_ evolution rate of 5.38 mmol·g^−1^·L^−1^. this rate is seven times higher than that of pure ZnIn_2_S_4_. Therefore, this work offers some helpful advice for the development of hollow p-n heterojunctions in the future and their utilization in the area of energy conversion.

## Figures and Tables

**Figure 1 nanomaterials-13-00758-f001:**
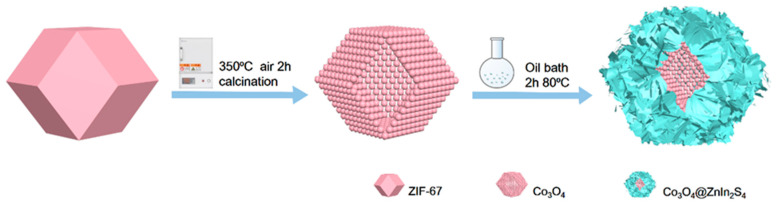
Scheme of the formation of the ZnIn_2_S_4_@Co_3_O_4_ p-n heterojunction.

**Figure 2 nanomaterials-13-00758-f002:**
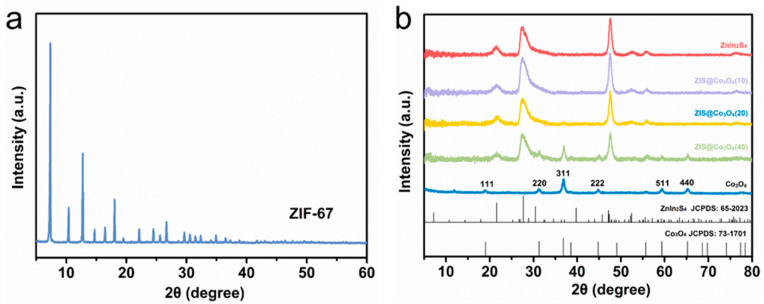
XRD spectra of all samples: (**a**) ZIF-67; (**b**) Co_3_O_4_, ZnIn_2_S_4_, Co_3_O_4_(10)@ZIS, Co_3_O_4_(20)@ZIS, Co_3_O_4_(40)@ZIS.

**Figure 3 nanomaterials-13-00758-f003:**
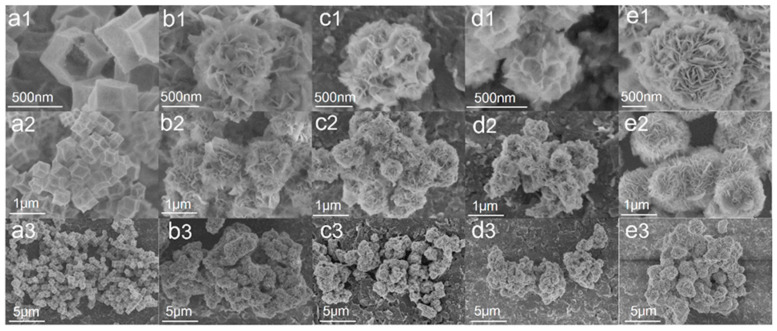
Field emission electron micrographs of all samples at different magnifications: (**a1**–**a3**) Co_3_O_4_; (**b1**–**b3**) Co_3_O_4_(10)@ZIS; (**c1**–**c3**) Co_3_O_4_(20)@ZIS; (**d1**–**d3**) Co_3_O_4_(40)@ZIS; (**e1**–**e3**) ZnIn_2_S_4_.

**Figure 4 nanomaterials-13-00758-f004:**
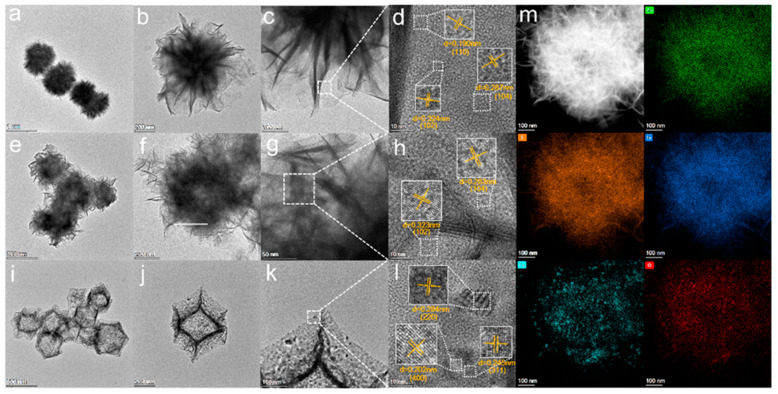
(**a**–**d**) STEM images of ZnIn_2_S_4_ and its HRTEM images; (**e**–**h**) STEM images of Co_3_O_4_(20)@ZIS and its HRTEM images; (**i**–**l**) STEM images of Co_3_O_4_ and its HRTEM images; (**m**) HAADF-STEM images of Co_3_O_4_(20)@ZIS and the corresponding elemental distribution images.

**Figure 5 nanomaterials-13-00758-f005:**
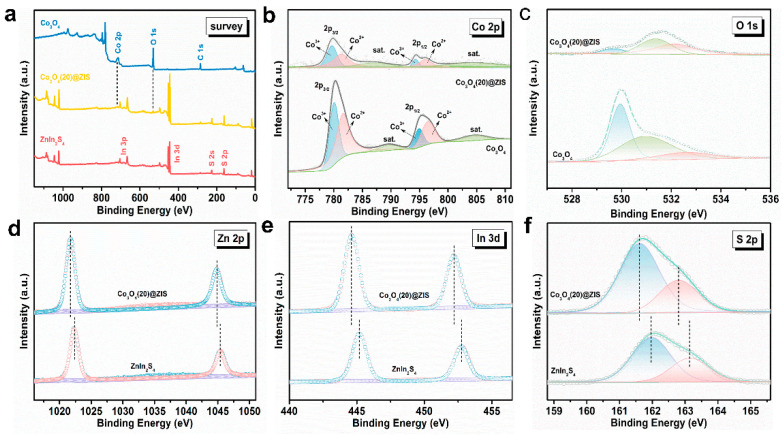
(**a**) General XPS images of Co_3_O_4_, Co_3_O_4_(20)@ZIS, and ZnIn_2_S_4_; (**b**) Co 2p;.(**c**) O 1s; (**d**) Zn 2p; (**e**) In 3d; (**f**) S 2p.

**Figure 6 nanomaterials-13-00758-f006:**
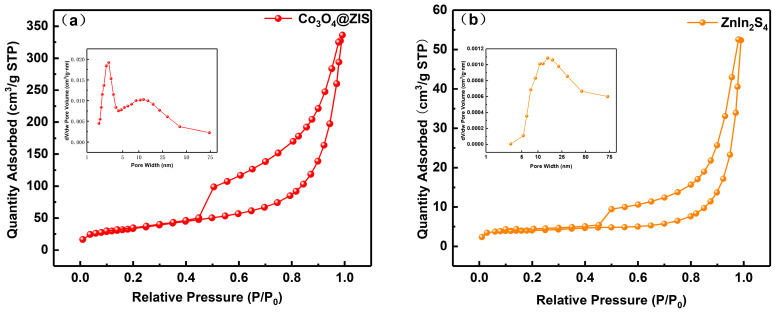
Nitrogen adsorption-desorption isotherms and corresponding pore size distribution curves of (**a**) Co_3_O_4_(20)@ZIS, (**b**) ZnIn_2_S_4_, and (**c**) Co_3_O_4_.

**Figure 7 nanomaterials-13-00758-f007:**
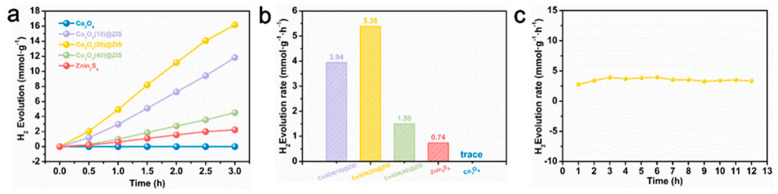
(**a**) Plot of total hydrogen production for each sample; (**b**) Plot of hydrogen production rate for each sample; (**c**) Long time hydrogen production rate of Co_3_O_4_(20)@ZIS.

**Figure 8 nanomaterials-13-00758-f008:**
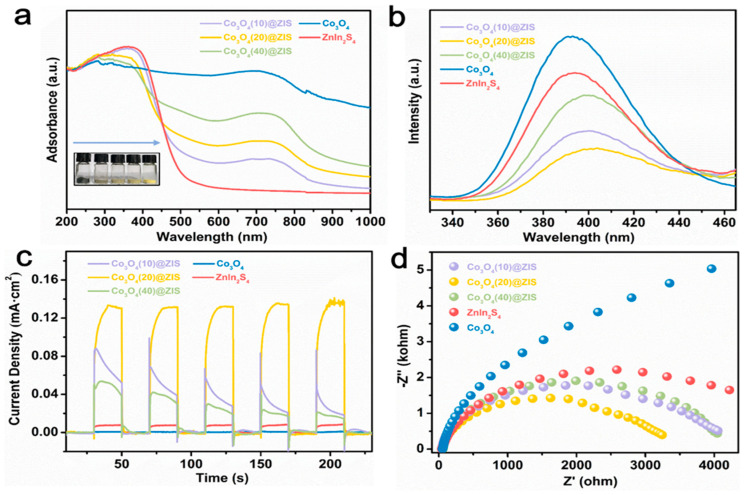
(**a**) DRS plots; (**b**) PL plots; (**c**) photocurrent data; (**d**) EIS data for all samples.

**Figure 9 nanomaterials-13-00758-f009:**
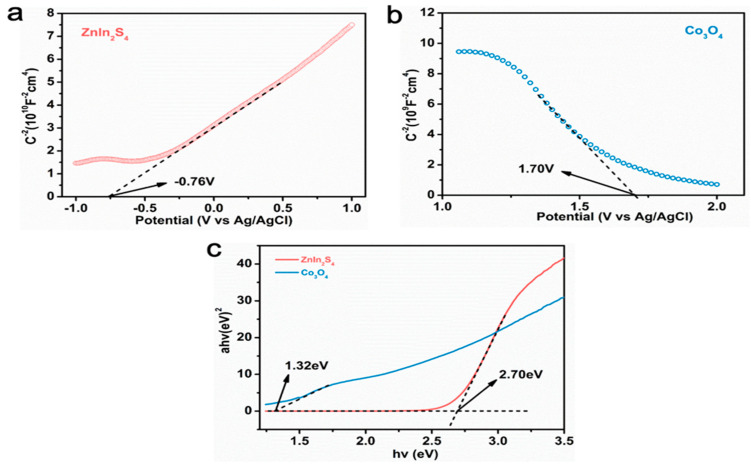
(**a**,**b**) The Mott-Schottky curve of ZnIn_2_S_4_ and Co_3_O_4_;(**c**) The band gap of ZnIn_2_S_4_ and Co_3_O_4_.

**Figure 10 nanomaterials-13-00758-f010:**
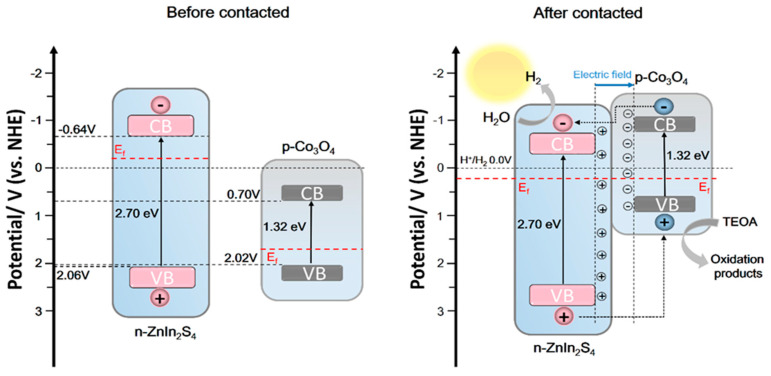
Mechanism of photocatalytic hydrogen production from Co_3_O_4_(X)@ZnIn_2_S_4_ p-n heterojunction.

**Table 1 nanomaterials-13-00758-t001:** Adsorption parameters of Co_3_O_4_(20)@ZIS, ZnIn_2_S_4_ and Co_3_O_4_.

Sample	S_BET_ (m^2^ g^−1^ )	Pore Volume (cm^3^ g^−1^ )	Average Pore Size (nm)
Co_3_O_4_(20)@ZIS	121	0.53	15.39
ZnIn_2_S_4_	15	0.08	31.15
Co_3_O_4_	71	0.35	19.03

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
