# Peer review of "Construction of Hollow Co3O4@ZnIn2S4 p-n Heterojunctions for Highly Efficient Photocatalytic Hydrogen Production"

_nanomaterials, 2023, doi:10.3390/nano13040758_

Round 1

Reviewer 1 Report

The article under analysis is well written and argued and presents the method of obtaining and characterization of ZnIn2S4@Co3O4 heterostructures showing excellent photocatalytic H2 activity. The hydrogen production rate is essentially constant within 12 hours.

The photocatalytic hydrogen production mechanism proposed in this work should be correlated to other data from the literature if they exist.

I also recommend the authors to include other techniques for characterizing the obtained heterostructures such as thermal analysis.

Author Response

Reviewer 1: The article under analysis is well written and argued and presents the method of obtaining and characterization of ZnIn2S4@Co3O4 heterostructures showing excellent photocatalytic H2 activity.  The hydrogen production rate is essentially constant within 12 hours.

Response: We greatly appreciate the reviewer’s positive comments on our work, and have made some revisions following the kind suggestions.

1.The photocatalytic hydrogen production mechanism proposed in this work should be correlated to other data from the literature if they exist.

Response: We greatly appreciate the reviewer’s valuable suggestion for improving the quality of our manuscript. The energy band gaps of Co3O4 and ZnIn2S4 are calculated to be 1.32 eV and 2.70 eV based on the optical absorption band. This result is helpful to the energy band mechanism. In the future, we'll endeavor to establish a connection.

  1. I also recommend the authors to include other techniques for characterizing the obtained heterostructures such as thermal analysis.

Response: We thank the reviewer for the time and effort to read and comment on our work. As suggested, TG and DSC information have been updated in the supporting information. TG analysis showed that the catalyst lost 5% weight from 0℃ to 1000℃, which indicated the stability of the Co3O4(20)@ZIS.

Reviewer 2 Report

This manuscript presents, Construction of Hollow Co3O4@ZnIn2S4 p-n Heterojunctions for Highly Efficient Photocatalytic Hydrogen Production. Authors successfully prepared Co3O4@ZIS p-n heterojunction and used as highly efficient photocatalytic hydrogen production. The manuscript is well written and presented well. However, following revisions should be made before publication:

1.      Authors should explain the formation of ZnIn2S4@Co3O4 I result and section.

2.      The importance of metal oxide and sulfide can be explained in the introduction section with the integrity of following article: doi.org/10.1016/j.compositesb.2022.110339, doi.org/10.1016/j.cplett.2022.139884

3.      The HR-TEM images and labeling in Figure 4 d, h and I should be clear.

4.      The EIS circuit diagram should be used to explain the parameters for the study of charge transfer resistance. Please refer: doi.org/10.1016/j.est.2021.103674

5.      How about the electrochemical H2 production of Co3O4@ZIS p-n heterojunction materials.

6.      The specific surface area and porosity of the materials is also a key parameter for the performance of materials. Therefore, BET and BJH information should be updated in the revised manuscript.

Author Response

Reviewer 2: This manuscript presents, Construction of Hollow Co3O4@ZnIn2S4 p-n Heterojunctions for Highly Efficient Photocatalytic Hydrogen Production. Authors successfully prepared Co3O4@ZIS p-n heterojunction and used as highly efficient photocatalytic hydrogen production. The manuscript is well written and presented well. However, following revisions should be made before publication:

Response: We greatly appreciate the reviewer’s positive comments on our work, and have made some revisions after carefully considering the suggestions and supplement more experimental studies.

  1. Authors should explain the formation of ZnIn2S4@Co3O4 I result and section.

Response: We thank the reviewer for insight and comment, and have revised the manuscript in order to clarify the questions provided.

At room temperature, Zn2+ and 2-methylimidazole synthesize regular dodecahedral ZIF-67 in methanol solution, which is subsequently placed in a muffle furnace for calcination. The organic ligand in ZIF-67 will undergo oxidation, forming hollow, regular dodecahedrons made of Co3O4 particles, and other gases will be liberated. The hollow regular dodecahedron Co3O4 will undergo a half-hour ultrasonic stir with sulfur, indium, zinc, and glycerol, followed by a two-hour immersion in oil at 80°C in a round-bottled flask. This results in the hollow Co3O4 surface homogeneous load with extremely thin ZnIn2S4 heterojunction material.

  1. The importance of metal oxide and sulfide can be explained in the introduction section with the integrity of following article: doi.org/10.1016/j.compositesb.2022.110339, doi.org/10.1016/j.cplett.2022.139884

Response: We greatly thank the reviewer for pointing to the incomplete review of important literatures. According to the comment, references mentioned by reviewer have been cited and discussed in revised manuscript for completing the introduction.  Many thanks for the valuable suggestions.

  1. The HR-TEM images and labeling in Figure 4 d, h and I should be clear.

Response:We thank the reviewers for their insight and comments, and have revised the manuscript to provide as much clarity as we can.

Figure 4. (a, b, c, d) Scanning transmission image of ZnIn2S4 and its HRTEM images; (e, f, g, h) Scanning transmission image of Co3O4(20)@ZIS and its HRTEM images; (i, j, k, l) Scanning transmission image of Co3O4 and its HRTEM images;.(m) HAADF-STEM images of Co3O4(20)@ZIS and the corresponding elemental distribution images.

  1. The EIS circuit diagram should be used to explain the parameters for the study of charge transfer resistance. Please refer: doi.org/10.1016/j.est.2021.103674

Response: According to the comment, references mentioned by reviewer have been cited and discussed in revised manuscript for completing the introduction. Thanks for the valuable suggestions. 

  1. How about the electrochemical H2production of Co3O4@ZIS p-n heterojunction materials?

Response: We greatly thank the reviewer for focusing on electrochemical H2 production of Co3O4@ZIS p-n heterojunction materials.

The optimal Co3O4 (20)@ ZnIn2S4 photocatalyst exhibited an H2 evolution rate of 5.38 mmol·g-1·L-1. this rate is 7 times higher than that of pure ZnIn2S4. Therefore, this work offers some helpful advice for the development of hollow p-n heterojunctions in the future and their utilization in the area of electrochemical H2 production.

  1. The specific surface area and porosity of the materials is also a key parameter for the performance of materials. Therefore, BET and BJH information should be updated in the revised manuscript.

Response: We thank the reviewer for the time and effort to read and comment on our work. As suggested, BET and BJH information have been updated in the revised manuscript.

The samples' nitrogen adsorption-desorption and pore size distribution curves are shown in Figure a–c. Naturally, type IV isotherm and type H3 hysteresis rings are found in all samples, demonstrating the presence of capillary condensation. Materials' microstructure and specific surface area are typically closely connected. As can be observed from the little figure, the three samples all have pore sizes that are mostly spread between 1 and 75 nm, which indicates that they are all mesoporous materials.

In Table 1, Co3O4(20)@ZIS, ZnIn2S4 and Co3O4 had specific surface area distributions of 121, 15 and 71 m2 g-1, respectively. Evidently, the specific surface area and pore volume of the Co3O4(20)@ZIS composite samples increased with the addition of ZnIn2S4 in comparison to pure Co3O4. It is possible that the composite catalyst will offer more active centers, which will make it easier for photocatalytic hydrogen evolution events to occur. Another angle is that the success of the catalyst preparation is demonstrated by the modification of the adsorption characteristics of the composite samples. 

Fig. 1 Nitrogen adsorption-desorption isotherms and corresponding pore size distribution curves of (a) Co3O4(20)@ZIS, (b) ZnIn2S4 and (c) Co3O4.

Table 1 Adsorption parameters of Co3O4(20)@ZIS, ZnIn2S4 and Co3O4.

Sample

SBET (m2 g-1 )

Pore volume (cm3 g-1 )

Average pore size (nm)

Co3O4(20)@ZIS

121

0.53

15.39

ZnIn2S4

15

0.08

31.15

Co3O4

71

0.35

19.03

Reviewer 3 Report

Results presented are classical complex investigations of new photocatalysts with heterojunctions for water splitting. Characterization and description of  photocatalytic and photoelectric properties are comprehensively presented. 

Results are original and finaly authors offer efficient composite photocatalytic system for hydrogen production. 

Manuscript could be published in the journal of Nanomaterials in present form.

Author Response

Reviewer 3: Results presented are classical complex investigations of new photocatalysts with heterojunctions for water splitting. Characterization and description of photocatalytic and photoelectric properties are comprehensively presented. 

Results are original and finally authors offer efficient composite photocatalytic system for hydrogen production. 

Manuscript could be published in the journal of Nanomaterials in present form.

Response: We greatly appreciate the reviewer’s positive comments on our work. After that, we will continue to explore the follow-up work of ZnIn2S4@Co3O4 p-n heterojunction.

Reviewer 4 Report

Manuscript deals with the Construction of Hollow Co3O4@ZnIn2S4 p-n Heterojunctions for Highly Efficient Photocatalytic Hydrogen Production, But publication point of view some modifications necessary.

1. There are grammatical mistakes. Please check the manuscript for grammar and English.

2. What is novelty of the present work? Rewrite it at the end of introduction section.

3. Compare your results with other researcher work in tabular form.

4. XRD or XPS should characterize Co3O4(20)@ZIS  after photocatalytic reaction to check the stability of the catalyst.

5. To enrich the literature in the manuscript add some literature.

 i. Catalysts 12 (10), 2022, 1185,  ii. Journal of Photochemistry and Photobiology A: Chemistry 434, 2023, 114250, iii.  Journal of Alloys and Compounds 928, 2022, 167133, iv. Materials Research Bulletin 133, 2021, 111026.

Author Response

Reviewer 2: Manuscript deals with the Construction of Hollow Co3O4@ZnIn2S4 p-n Heterojunctions for Highly Efficient Photocatalytic Hydrogen Production, But publication point of view some modifications necessary.

Response: We greatly appreciate the reviewer’s positive comments on our work, and have made some revisions after carefully considering the suggestion.

  1. There are grammatical mistakes. Please check the manuscript for grammar and English.

Response: We thank the reviewer for the time and effort to read and comment on our work, and have revised the manuscript. We have checked the manuscript again and corrected the grammatical errors. Thank you for your careful comments.

  1. What is novelty of the present work? Rewrite it at the end of introduction section.

Response: We thank the reviewer for the time and effort to read and comment on our work. 

Herein, we successfully fabricated ZnIn2S4@Co3O4 heterojunction by loading ultra-thin two-dimensional ZnIn2S4 nanosheets onto hollow dodecahedral Co3O4 nanocages derived from ZIF-67 by oil bath. The constructed ZnIn2S4@Co3O4 p-n heterojunction effectively hinders electron-hole recombination. Additionally, the dark hollow structure of Co3O4 significantly expands the spectrum of light absorption and the heterojunction's ability to use light efficiently by preventing the accumulation of ZnIn2S4 nanosheets. Consequently, the optimal ZnIn2S4@Co3O4 composites show apparent enhancement in H2 production efficiency compared with pure ZnIn2S4. The ZnIn2S4@Co3O4 heterostructure is stable within 12h.

  1. Compare your results with other researcher work in tabular form.

Response: We greatly appreciate the reviewer’s valuable suggestion for improving the quality of our manuscript. Following the suggestions, we add a table to Compare your results with other researcher work. We put this table in the supporting information.

Semiconductor

Solvothermal

method

Morphology

Sacrificial reagent

Light source

(nm)

Optimal hydrogen

evolution

(μmol·g1·h1)

Refs.

ZnIn2S4@

NH2-MIL-

125(Ti)

Solvothermal

method

Micropores

and mesopores

Na2S/Na2SO3

300 W Xe

lamp

λ > 420 nm

2204.2 40 wt%

NH2-MIL-125(Ti)

1

g-C3N4@

ZnIn2S4

Solvothermal

method

2D/2D g-C3N4

nanosheet@

ZnIn2S4

nanoleaf

Triethanolamine

300 W Xe

lamp

λ > 420 nm

2780

2

Co(dmgH)2pyC

/ZnIn2S4

Impregnating

method

Microspheres

Triethanolamine

300 W Xe

lamp

λ > 420 nm

3840

3.0 wt%

Co(dmgH)2pyCl/

ZnIn2S4

3

CQDs/ZnIn2S4

Microwave

hydrothermal

method

Microspheres

N/A

350 W Xe

lamp

λ > 400 nm

1032.2

4

MoS2/ZnIn2S4

Hydrothermal

method

Nanoparticles

Na2S/Na2SO3

300 W Xe

lamp

λ > 420 nm

2080

0.5 wt%

MoS2/ZnIn2S4

5

In2S3/ZnIn2S4

Ion-exchange

method

Microflowers

Na2S/K2SO3

300 W Xe

lamp

λ ≥ 420 nm

678

6

CdS/ZnFe2O4/

ZnIn2S4

Solvothermal

processes & Ionic

layer adsorption-

reaction method

Nanosheet

stereoscopic

films

Na2S/Na2SO3

300 W Xe

lamp

λ > 420 nm

79.0 μmol h−1

1-CdS/ZnFe2O4/

ZnIn2S4

7

[1] H. Liu, J. Zhang, D. Ao, Construction of heterostructured ZnIn2S4@NH2-MIL-

125(Ti) nanocomposites for visible-light-driven H2 production, Appl. Catal. B

Environ. 221 (2018) 433–442.

[2] B. Lin, H. Li, H. An, W. Hao, J. Wei, Y. Dai, C. Ma, G. Yang, Preparation of 2D/2D

g-C3N4 nanosheet@ZnIn2S4 nanoleaf heterojunctions with well-designed high-

speed charge transfer nanochannels towards high-efficiency photocatalytic hy-

drogen evolution, Appl. Catal. B Environ. 220 (2018) 542–552.

[3] Y. Gao, H. Lin, S. Zhang, Z. Li, Co(dmgH)2pyCl as a noble-metal-free co-catalyst for

highly efficient photocatalytic hydrogen evolution over hexagonal ZnIn2S4, RSC

Adv. 6 (2016) 6072–6076.

[4] Q. Li, C. Cui, H. Meng, J. Yu, Visible-light photocatalytic hydrogen production

activity of ZnIn2S4 microspheres using carbon quantum dots and platinum as dual

co-catalysts, Chem. Asian. J. 9 (2014) 1766–1770.

[5] T. Huang, W. Chen, T. Liu, Q. Hao, X. Liu, Znln2S4 hybrid with MoS2: a non-noble

metal photocatalyst with efficient photocatalytic activity for hydrogen evolution,

Powder. Technol. 315 (2017) 157–162.

[6] Z. Mei, S. Ouyang, D. Tang, T. Kako, D. Golberg, J. Ye, An ion-exchange route for

the synthesis of hierarchical In2S3/ZnIn2S4 bulk composite and its photocatalytic

activity under visible-light irradiation, Dalton Trans. 42 (2013) 2687–2690.

[7] Y. Chen, G. Tian, W. Zhou, Y. Xiao, J. Wang, X. Zhang, H. Fu, Enhanced photo-

generated carrier separation in CdS quantum dot sensitized ZnFe2O4/ZnIn2S4

nanosheet stereoscopic films for exceptional visible light photocatalytic H2 evo-

lution performance, Nanoscale 9 (2017) 5912–5921.

  1. XRD or XPS should characterize Co3O4(20)@ZIS  after photocatalytic reaction to check the stability of the catalyst.

Response: We thank the reviewer for concerning the thermal stability of the catalyst.

In order to prove the stability of the catalyst, we added three cycles of experiments, and the hydrogen production decrease from 16.16 mmol·g-1 to 14.77 mmol·g-1 in three hours. We put this table in the supporting information.

  1. To enrich the literaturein the manuscript add some literature.

Response: According to the comment, references mentioned by reviewer have been cited and discussed in revised manuscript for completing the introduction. Thanks for the valuable suggestions.

Round 2

Reviewer 4 Report

The revision made by the author is satisfactory. The present form of a manuscript should be accepted.